A simple and efficient cloning system for CRISPR/Cas9-mediated genome editing in rice

Liu Xiaoli 1
Zhou Xiujuan 1
Li Kang 2
Wang Dehong 2
Ding Yuanhao 1
http://orcid.org/0000-0002-3659-9110 Liu Xianqing 1
http://orcid.org/0000-0002-3659-9110 Luo Jie 1 2
http://orcid.org/0000-0003-3588-8743 Fang Chuanying 1 cyfang@hainanu.edu.cn
1 College of Tropical Crops, Hainan University , Haikou, Hainan , China
2 National Key Laboratory of Crop Genetic Improvement and National Centre of Plant Gene Research, Huazhong Agricultural University , Wuhan, Hubei , China
Lu Kun
Electronic publication date: 2020 Jan 29
Publication date: 2020
Volume: 8
Electronic Location ID: e8491
Received 2019 Oct 22; Accepted 2019 Dec 30
Copyright: © 2020 Liu et al.
Copyright year: 2020
Copyright holder: Liu et al.
License: This is an open access article distributed under the terms of the Creative Commons Attribution License, which permits unrestricted use, distribution, reproduction and adaptation in any medium and for any purpose provided that it is properly attributed. For attribution, the original author(s), title, publication source (PeerJ) and either DOI or URL of the article must be cited.
License URL: https://creativecommons.org/licenses/by/4.0/

Keywords: Cloning system, Multiplex PCR, CRISPR/Cas9, Genome editing, Rice

Funding: National Natural Science Foundation of China 31800250 and 31960063 Hainan University Startup Fund KYQD(ZR)1824 This work was supported by the National Natural Science Foundation of China (Nos. 31800250 and 31960063), and the Hainan University Startup Fund (No. KYQD(ZR)1824). The funders had no role in study design, data collection and analysis, decision to publish, or preparation of the manuscript.

==============================
Rapidly growing genetics and bioinformatics studies provide us with an opportunity to obtain a global view of the genetic basis of traits, but also give a challenge to the function validation of candidate genes. CRISPR/Cas9 is an emerging and efficient tool for genome editing. To construct expression clones for the CRISPR/Cas9, most current methods depend on traditional cloning using Gateway reaction or specific type IIS restriction enzymes and DNA ligation, based on multiple steps of PCR. We developed a system for introducing sgRNA expression cassette(s) directly into plant binary vectors in one step. In this system, one sgRNA expression cassette(s) is generated by an optimized multiplex PCR, in which an overlapping PCR took place. Whilst, two sgRNA expression cassettes were amplified in a single round of PCR. Subsequently, an LR or Golden gate reaction was set up with unpurified PCR product and befitting destination vector. We are able to construct expression clones within 36 h, which greatly improves efficiency and saves cost. Furthermore, the efficiency of this system was verified by an agrobacterium-mediated genetic transformation in rice. The system reported here provides a much more efficient and simpler procedure to construct expression clones for CRISPR/Cas9-mediated genome editing.

Introduction

Genetic mutants are essential for understanding gene functions in both basic and applied research. In the past decades, many important biological mechanisms have been revealed with the application of mutant libraries (Fang et al., 2018; Li et al., 2018). Mutant libraries of model plants tend to be generated by physical, chemical, or biological (e.g., T-DNA/transposon insertion) mutagenesis, resulting in random mutagenesis (Jeon et al., 2000; Till et al., 2007). However, mutant libraries may not cover candidate genes or sometimes carried undesirable mutations. Hence, targeted genome editing is critical for generation of mutants to meet the demands of gene function analyses. Programmable sequence-specific nucleases (SSNs) can introduce double-stranded breaks (DSB) in specific chromosomal sites. Mutants are then generated by either imprecise repair of DSBs mediated by the error-prone non-homologous end joining pathway or by insertion of donor templates into the break by homology-directed repair (Symington & Gautier, 2011). Several nucleases have been documented to producing SSNs, such as the zinc finger nucleases (Miller et al., 2007; Sander et al., 2011) and the transcription activator-like effector nucleases (Bogdanove & Voytas, 2011; Streubel et al., 2012). The abovementioned nucleases enable us to edit plant genome (Li et al., 2012; Shukla et al., 2009; Zhang et al., 2010). Recently, the bacteria type II clustered regularly interspaced short palindromic repeat (CRISPR) and CRISPR associated (Cas) protein system is emerging as a promising tool for targeted genome editing. The Cas9 endonuclease from Streptococcus pyogenes, coupled with a sequence-specific single guide RNA (sgRNA), is able to target the DNA sequence of 5′-N(20)NGG, in which N20 is the same as 20 bases of the sgRNA 5′ sequence (referred to as sgRNA spacer hereafter) and NGG is the protospacer-adjacent motif (PAM) (Jinek et al., 2012; Mali et al., 2013; Shalem, Sanjana & Zhang, 2015).

The CRISPR/Cas9 technology was applied in several plant species, such as Arabidopsis, rice, maize, cotton and wheat (Govindan & Ramalingam, 2016; He et al., 2018; Lowder, Malzahn & Qi, 2016; Wang et al., 2018; Yin, Gao & Qiu, 2017). Many efforts have been made to accelerate the usage of the CRISPR-Cas9 system in rice genome editing. Recently, by which the Cas9-sgRNA complex recognizes its target DNA has been elucidated by the structural study. The critical role of PAM interacting domain of Cas9 in the recognition of 5′-NGG-3′ PAM has been revealed by functional analyses (Nishimasu et al., 2017). Based on these mechanisms, variants of Cas9 has been developed to cleave sites containing altered PAM in rice, which greatly expanded the range of genome editing (Hu et al., 2016). Furthermore, new technologies have been developed for gene activation (Li et al., 2017), replacement and insertion (Li et al., 2016), which satisfy various demands of genome editing for functional genomic studies. Moreover, the efficacy of the CRISPR-Cas9 system has been demonstrated in the construction of genome-wide mutant libraries in rice (Lu et al., 2017; Meng et al., 2017).

The sgRNAs with a target sequence (target-sgRNAs) and Cas9 protein are essential for CRISPR/Cas9-mediated genome editing. Gateway binary vectors for co-expression of Cas9 and sgRNA have been developed to accommodate this system to Agrobacterium-mediated transformation. A gene-specific sgRNA spacer is inserted into an entry clone (pOs-sgRNA) to express a target-sgRNA by a restriction-ligation reaction. Subsequently, the target-sgRNA cassette is cloned into destination clone (pH-Ubi-cas9-7) by LR reaction, containing Cas9 driven by the maize ubiquitin promoter. This system performed high efficiency in rice genome editing (Feng et al., 2013). Furthermore, a CRISPR/Cas9 system for multiplex genome editing in rice has been established. In this system, a gene-specific spacer is cloned into intermediate vectors by a restriction–ligation reaction to produce sgRNA expression cassette. Subsequently, adaptors for further exonuclease-based Gibson Assembly or Type IIS restriction enzymes-based Golden Gate cloning are fuzed to the sgRNA expression cassette by PCR amplification. Alternatively, the sgRNA expression cassette with a specific spacer and adaptors could be obtained by overlapping PCR. Then the sgRNA expression cassette is introduced to the binary vector carrying Cas9 with Golden Gate cloning or Gibson Assembly, respectively. To introduce a single or multiple sgRNA expression cassette(s) into the CRISPR/Cas9 binary vectors, Golden Gate cloning or Gibson Assembly is carried out. Besides, OsU3 and OsU6 promoters were used for expressing multiple sgRNA cassettes with definite transcription initiation sites (Ma et al., 2016). Collectively, traditional cloning with restriction enzymes and DNA ligase and overlapping PCR with two rounds of amplifications are most methods to obtain binary vectors. Notwithstanding the wide-use of the abovementioned systems, the construction of binary vectors containing Cas9 and target-sgRNA cassettes is time-consuming.

Rapid growing population genetics and bioinformatics provide us with a chance to obtain a global view of the genetic basis of a specific trait of plants (Fang, Fernie & Luo, 2019; Fang & Luo, 2019). For example, a single study could identify dozens of genes that may modulate levels of metabolites (Chen et al., 2014, 2016; Fang et al., 2016; Zhu et al., 2018). However, it is also challenging for us to rapidly and efficiently confirm the functions of numerous candidate genes. Hence, a more efficient system to construct vectors for genome editing is of a vital role. Herein, we reported a novel method to construct the binary vectors with one or two targets by a single round of PCR amplification and a single LR reaction or Golden Gate cloning.

Materials and Methods

Primers design

Primers for amplification were designed using Oligo 7 (http://www.oligo.net/) at default settings with primer melting temperature at 58 °C ± 3 °C. In Gateway clone, the 5′-end of the forward and reverse primers were tagged with aaaaagcaggctta (attB1) and agaaagctgggta (attB2) respectively.

Sequences of rice genes were obtained from the Michigan State University Rice Genome Annotation Project database (Kawahara et al., 2013) (http://rice.plantbiology.msu.edu). Targets in exons of each gene for genome editing by CRISPR/Cas9 were selected using CRISPR-P tools (Lei et al., 2014). All the primers used in this study are listed in Table S1, in which adapters are shown in lower case and restriction sites are shown in lower case and italic.

Construction of donor and destination vectors

Given that there is an EcoRV-cutting site on the backbone of pDONR207, a donor vector was constructed by inserting a spacer sequence containing an EcoRV-cutting site between the OsU3 promoter and sgRNA (Fig. 1A). The spacer and its reverse complement were fuzed to the 3′ downstream of sgF primer (denoted as OJG109) and U3R primer (denoted as OJG110), respectively (Table S1). The gateway system matched attB adaptors were fuzed to the 5′ upstream of U3F and sgR primer (denoted as OJD387 and OJD388, Table S1). Then an overlapping PCR was performed with two rounds of reactions: (i) OsU3 promoter fuzed with the spacer was amplified by OJD387 and OJG110, with pOs-sgRNA (Feng et al., 2013) as the template; the spacer fuzed with sgRNA was amplified by OJD388 and OJG109, with pOs-sgRNA (Feng et al., 2013) as the template; (ii) the product of the first round of PCR were equally proportion mixed and diluted to 1/100 as the template of the second round of PCR using OJD387 and OJD388 as primers (Table S1). Subsequently, an BP reaction was set up with 0.4 ul of PCR product and 40 ng of pDONR207 (Fig. 1B). After incubation for 3 h under 25 °C, 1 ul of the product was introduced into 20 ul of T1 competent cells (ZOMANBIO Co., Shanghai, China). Positive clones were identified by colony PCR. Then the plasmid was verified by DNA sequencing and renamed as PJF997 (Fig. 1C). The donor vector named PJF999 was constructed under a similar procedure with PJF997, carrying sgRNA driven by OsU6a. The first round PCR was set up with pYLsgRNA-OsU6a (Ma et al., 2016) as the template (Fig. 1).

Figure 1 Construction of the donor vectors as the templates for multiplex PCR amplification of target-sgRNA expression cassettes.

(A) A spacer sequence containing an EcoRV-cutting site was designed. (B) The spacer sequence (AGATATCGAGAGGGATGGGG) and its reverse complementary sequence (CCCCATCCCTCTCGATATCT) were added to the 5′-end of the forward primer of sgRNA (sgF) and reverse primer of OsU3/U6a promoter (U3R/U6R), respectively. The resulting primers were denoted as OJG109 (new sgF), OJG110 (new U3R), OJG112 (new U6R). The forward primer of OsU3/U6a promoter (U3F/U6F) and the reverse primer of sgRNA (sgR) were tagged with attB1 and attB2 adapter at the upstream, respectively, which were denoted as OJD387 (new U3F), OJD389 (new U6F) and OJD388 (new sgR). An overlapping PCR was performed with two round of reactions: (i) with pOs-sgRNA as the template, PCR reactions were set up with OJD387 and OJG110 or OJD388 and OJG109; (ii) the product of the first round of PCR were equally proportion mixed and diluted to 1/100 as the template of the second round of PCR using OJD387 and OJD388 as primers. Subsequently, an BP reaction was set up with 0.4 ul of PCR product and 40 ng of pDONR207. After incubation for 3 h under 25 °C, one ul of the product was introduced into 20 ul of T1 competent cells (ZOMANBIO Co., Shanghai, China). Positive clones were sequenced and renamed as PJF997, which can be further used as template for multiplex PCR amplification of target-sgRNA expression cassettes driven by OsU3. The donor vector PJF999 as template for multiplex PCR amplification of target-sgRNA expression cassettes driven by OsU6a was constructed as the same workflow, using pYLsgRNA-OsU6a (Ma et al., 2016) as the template. (C) The schematic diagram of PJF997 and PJF999.

Overlapping PCR was performed to produce the cassette of sgRNA-OsU6a promoter. Primers named OJP052 and OJG646 were used to amplify OsU6a containing attB adaptor at the downstream and overlapped sequence with sgRNA at the upstream. Similarly, sgRNA fuzed with attB adaptors at the upstream was obtained by PCR reaction with primers named OJG645 and OJP051. Overlapping PCR was then set up with the abovementioned products and attB adaptor primers (OJP051 and OJP052). The PCR product was then introduced into pDONR207 with a BP reaction, producing the plasmid PJG090 as the template for PCR amplification of two spacers (Fig. S1).

There are two BsaI (a widely used enzyme for Golden Gate cloning) sites in the backbone of pH-Ubi-cas9-7, limiting its application in Golden Gate reaction with BsaI. To remove the BsaI sites, primers containing site mutation in its recognition site was designed (referred to as OJK121 and OJK122). The BsaI sites and matched adaptors were added to the 5′ stream of primers. An amplification with OJK121 and OJK122 was conducted with pH-Ubi-cas9-7 as the template. Gel-purified PCR product and pH-Ubi-cas9-7 was then digested with BsaI and then submitted to a ligation reaction. The product was then transferred into DB3.1 competent cells (ZOMANBIO Co., Shanghai, China). Positive clones were then verified by DNA sequencing and digestion with BsaI. The new vector removing BsaI sites was renamed as PJG097. Subsequently, an LR reaction was performed with 30 ng pOs-sgRNA and PJG097, producing the destination clone named PJG112 (Fig. 2).

Figure 2 Construction of the destination vectors PJG097 without BasI sites.

There are two BsaI sites in the backbone of pH-Ubi-cas9-7. To remove the BsaI sites, primers containing site mutation in its recognition site was designed (referred to as OJK121 and OJK122). The BsaI sites and matched adaptors were added to the 5′ stream of primers. An amplification with OJK121 and OJK122 was conducted with pH-Ubi-cas9-7 as the template. Gel-purified PCR product and pH-Ubi-cas9-7 was then digested with BsaI and then submitted to a ligation reaction. The product was then transferred into DB3.1 competent cells (ZOMANBIO Co., Shanghai, China). Positive clones were then verified by DNA sequencing and digestion with BsaI. The new vector removing BsaI sites was renamed as PJG097. An LR reaction was performed with 30 ng pOs-sgRNA and PJG097, producing the destination clone named PJG112. PJG112 could serve as destination vector to construct expression clones containing two spacers.

Construction of expression clones

Two fragments were produced by digestion of PJF997 and PJF999 with EcoRV, which were gel-purified into one cube and used as PCR templates for amplification of target-sgRNA cassettes driven by OsU3 and OsU6a, respectively.

To obtain cassettes containing attL sites in the PCR products, OJP001 and OJP002 (Table S1) were used as universal primers for digested PJF997 and PJF999. A 20-bp length region starting with A nucleotide followed by a PAM site on the exon of OsCCD8 was selected as a targeted spacer. The spacer and its reverse complement were fuzed to the 3′ downstream of sgF and U3R primers, producing spacer primers named OJD383 and OJD384. The overlapping was accomplished in a single round of optimized multiplex PCR with KOD FX. The PCR was set up with about 0.1 ng/ul of digested PJF997, universal primers (300 nM each) and spacer primers (20 nM each). After amplification with 35 cycles, 0.4 ul of PCR product was used directly in the LR reaction with 40 ng of PJG097. After incubation under 25 °C for 3 h, 1 ul of the product was introduced into 20 ul of T1 competent cells. Positive clones were identified by colony PCR. Then the expression vector for CRISPR/Cas9 mediated genome editing was verified by DNA sequencing and renamed as PJD392.

The procedures for the construction of expression vector for CRISPR/Cas9 mediated genome editing of OsDWARF14 resembled that for OsCCD8. A 20-bp length region starting with G nucleotide followed by a PAM site on the exon of OsDWARF14 was selected as a targeted spacer. Spacer primers named OJG521 and OJG522 (Table S1) were used in the amplification of sgRNA expression cassettes with digested PJF999 as the template. Finally, the expression clone named PJF625 was obtained.

Two spacers starting with A or G nucleotide were selected from the genome of OsRCD1. Primers containing adaptors for Golden Gate cloning (OJH307 and OJH308, Table S1) were used in the amplification with PJG090 as the template. The reagents were recommended as following: one ul of PCR product, 50 ng of PJG112, 1 ul of Cutsmart Buffer (NEB), 0.4 ul of T4 ligase buffer (NEB), 5 U of BsaI (NEB), 20 U of T4 DNA ligase (NEB) and add ddH2O to 10 ul. The reaction was incubated for 20–25 cycles (37 °C 2 min, 20 °C 5 min), followed by incubation at 50 °C and 80 °C for 5 min, respectively. Subsequently, 1 ul of the product was introduced into T1 competent cells. Positive clones were identified by colony PCR. Then the expression vector for CRISPR/Cas9 mediated genome editing was verified by DNA sequencing and renamed as PJF943.

Plant transformation

Transgenic plants were derived from Oryza sativa ssp japonica cv Zhonghua 11 (ZH11). The constructs were introduced into Agrobacterium tumefaciens strain EHA105 and then transferred into ZH11 as described previously (Hiei et al., 1994).

Genotype analysis of transgenic plants

The genomic DNA of transgenic plants was extracted using the sodium dodecyl sulfate method. T-DNA insertion was verified by PCR with primers named OJD393 and OJP039 (Table S1). For positive transgenic plants, genomic regions covering targeted spacers were amplified and sequenced. Mutation analysis was performed with the Degenerate Sequence Decode (DSDecode) program (Liu et al., 2015; Ma et al., 2015).

Results and Discussion

Construction of donor and destination vectors

Overlapping PCR is an efficient way to produce target-sgRNA expression cassettes, using a circular plasmid as the template. Traditionally, this method frequently produces undesirable amplification of the original plasmid sequence. We assumed that cutting OsU3/OsU6a promoter and sgRNA into two fragments could efficiently avoid undesirable amplifications. Hence, we set out to design new plasmids as templates for overlapping PCR amplification of sgRNA expression cassettes. Sequence analysis revealed that there is a recognition site of EcoRV on the backbone of pDONR207. An EcoRV-site between OsU3/OsU6a promoters and sgRNA could enable us to linearize template vectors by a single digestion reaction. Simply, template vectors were constructed by overlapping PCR and BP reactions. Firstly, a spacer sequence containing a recognition site of EcoRV was selected and sgRNA expression cassette with attB adaptors was produced by overlapping PCR amplification. Subsequently, a BP reaction was carried out with the PCR product and Gateway donor clone pDONR207. According to the above procedures, template vectors for OsU3- and OsU6a-driven sgRNA expression cassettes were generated, referred to as PJF998 and PJF999, respectively (Fig. 1).

Plasmids for two targets of genome editing containing the main elements in such order: OsU3 promoter, the first spacer, sgRNA, OsU6 promoter, the second spacer, sgRNA. Hence, the construction could be simplified by amplification the sequence between two spacers, followed by ligation into a binary vector containing the OsU3 promoter and sgRNA. Hence, overlapping PCR was performed to produce the sequence of sgRNA–OsU6a promoter. The PCR product with attB adaptors was then introduced into pDONR207 with a BP reaction, producing the plasmid PJG090 as the template for PCR amplification of two spacers. Previously reported pOs-sgRNA contains two recognition sites of BsaI between OsU3 promoter and sgRNA. Hence, the amplified product of two sgRNA expression cassettes could be introduced into pOs-sgRNA by Golden Gate cloning and then into the binary vector pH-Ubi-cas9-7 by an LR reaction. For the more efficient construction of binary vectors, we set out to modify the pH-Ubi-cas9-7 vector. Firstly, two recognition sites of BsaI on the backbone of pH-Ubi-cas9-7 were removed by point mutation, producing a binary vector as the substitutes of pH-Ubi-cas9-7 named PJG097. Subsequently, an LR reaction was carried out with PJG097 and pOs-sgRNA, producing in the plasmid named PJG112, which could serve as a binary vector for Golden Gate reaction with BsaI (Fig. 2).

Introducing sgRNA expression cassette(s) into the binary vector

We set out to construct expression clones for genome editing of OsDWARF14 and OsCCD8, whose mutants displayed dwarf and increased tillering in rice (Arite et al., 2007, 2009). A 20-bp length sequence from an exon of each gene was selected as the target, starting with A or G nucleotide for amplification of sgRNA expression cassettes containing OsU3 or OsU6a promoter, respectively. The sgRNA expression cassettes were easy to be obtained by overlapping PCR consists of two rounds of amplification. For the purpose of saving time and money, we set out to explore whether the overlapping PCR could be accomplished in a single round of amplification. Hence, a multiplex PCR reaction was performed with two pairs of primers. However, there were two fragments amplified without the targeted overlapped fragment (Fig. S2). Concentration and dosage of spacer primers were used to optimize the multiplex PCR (Fig. S2). Finally, 20 nM spacer primers were selected, for the high amplification efficiency and none undesirable fragments. The overlapped product containing attL sites was subjected to LR reaction with PJG097. Finally, expression clones for CRISPR/Cas9 mediated genome editing of OsDWARF14 and OsCCD8 were obtained, named PJF625 and PJD392, respectively (Fig. 3). To further reduce cost, the LR reaction was scaled down by 1/5 to 1/10 of the recommended reaction system. Introducing only 1 ul LR reaction production into 20 ul T1 competent cells is sufficient to obtain more than 100 clones.

Figure 3 Workflow for constructing expression clone containing a single target-sgRNA expression cassette with multiplex PCR.

Optimized multiplex PCR was set up with 300 nm of universal primers (OJP051 and OJP052) and 20 nm of spacer primers (SF and SR). After amplification with 35 cycles, 0.4 ul of PCR product was used directly in the LR reaction with 40 ng of PJG097. After incubation under 25 °C for 3 h, one ul of the product was introduced into 20 ul of Trans T1 competent cells. Positive clones were identified by clone PCR and sequenced.

To test the efficiency of the system, we set out to construct expression clones for genome editing of RCD1 with two spacers, with a 382 bp length interval between two theoretical editing sites (4 bp upstream of PAMs). Two spacers starting with A or G were selected. A single round of PCR was performed with PJG090 as the template, producing sgRNA-OsU6a sequences fuzed with two spacers. Subsequently, a Golden Gate cloning reaction was performed to introduce the product into PJG112. Finally, an expression clone named PJF943 for CRISPR/Cas9 mediated genome editing of OsRCD1 was produced (Fig. 4). Golden Gate cloning reactions with unpurified PCR products could yield about 10 clones with a high positive rate. In contrast, we were able to obtain more than 200 clones by Golden Gate cloning using purified fragments.

Figure 4 Workflow for constructing expression clone containing two target-sgRNA expression cassettes with Golden Gate clone.

Primers containing adaptors for Golden Gate cloning (OJH307 and OJH308) were used in the amplification with PJG090 as the template. The reagents were recommended as following: one ul of PCR product, 50 ng of PJG112, one ul of Cutsmart Buffer (NEB), 0.4 ul of T4 ligase buffer (NEB), 5 U of BsaI (NEB), 20 U of T4 DNA ligase (NEB) and add ddH2O to 10 ul. The reaction was incubated for 20–25 cycles (37 °C 2 min, 20 °C 5 min), followed by incubation at 50 °C and 80 °C for 5 min, respectively. Subsequently, one ul of the product was introduced into Trans T1 competent cells. Positive clones were identified by clone PCR and sequenced.

Practically, no more than three clones were verified by DNA sequencing. Almost all of the clones yielded by both methods carried desirable sgRNA expression cassette. Based on this system, we were able to obtain expression clones for CRISPR/Cas9 within 36 h. Compared to the previously reported restriction enzyme- and ligation-independent strategy (Khan et al., 2017), this system greatly increases the efficiency and saves the cost, because of the optimized templates for amplification of sgRNAs and/or improved multiplex PCR method.

In vivo verification of the expression vectors

To test whether the system is efficient for genome editing in rice, transgenic assays were performed with the abovementioned three expression vectors. Totally, 73 and 69 transgenic plants were produced by the agrobacterium-mediated genetic transformation of PJF625 and PJD392. The genotype of the offsprings was identified by DNA sequencing and analyzed with the DSDecode program. There were 36 and 35 plants carrying mutations in two alleles of OsDWARF14 and OsCCD8, respectively. Moreover, 19 and 15 heterozygous mutants were produced for OsDWARF14 and OsCCD8, respectively (Figs. 5A–5D; Table S2). All the mutants of OsDWARF14 and OsCCD8 displayed dwarf and increased tillering number (Fig. 5E), consistent with previous studies. Among the offsprings of PJF943, only 10 out of 92 plants harbor no mutation in both targeted regions of OsRCD1 (Table S2). Furthermore, a total of 36 plants carried both mutations in two sites, including two plants displayed a 382 bp-length deletion between targeted spacers (Fig. 5F), which is consistent with the estimated size. Collectively, more than 60% of the T0 plants carried a mutation in the targeted region, most of which were insertion with a single A 3-bp upstream of PAMs.

Figure 5 In vivo verification of the expression vectors.

The constructs were introduced into Agrobacterium tumefaciensstrain EHA105 and then transferred into ZH11 (WT). Representative Sanger sequencing chromatograms for negative transgenic lines of PJD392 (OsCCD8) (A) and PJF625 (OsDWARF14) (B). Chromatograms of mutants of OsCCD8 (C) and OsDWARF4 (D). (E) Phenotype of positive transgenic lines of PJD392, PJF625 and ZH11 (WT). White bar represents 20 cm. (F) PCR analysis of 15 selected T0 transgenic rice of PJF943 and WT. Red arrows represents two plant with long region deletion.

Conclusion

The CRISPR/Cas9 technology was applied in several plant species, which contributes to both basic research and plant breeding. Rapid growing population genetics and bioinformatics provide us with a chance to obtain a global view of the genetic basis of a specific trait of plants. However, it is also challenging for us to rapidly and efficiently confirm the functions of numerous candidate genes. Herein, we reported a novel method to construct the binary vectors with one or two targets by a single round of PCR amplification and a single LR reaction or Golden Gate cloning. With an optimized concentration of four primers in one tube, an expression cassette with one target is produced by an overlapping PCR, then subjected to an LR reaction. Compared with traditional methods, a multiplex PCR took place in a single round of PCR instead of two rounds of overlapping PCR. Moreover, an LR reaction was able to produce the binary vector, since the PCR products contain attL sites. By improving the template of PCR reactions, we are able to obtain PCR products with two targets and sgRNA–OsU6 promoter, which fits the Golden Gate reaction with a destination vector. Using this system, we are able to construct an expression clone within 36 h, which greatly improve efficiency and save cost.

Supplemental Information

Supplemental Information 1 Primers used in this study.

Click here for additional data file.

Supplemental Information 2 Summary of detected mutations in T0 transgenic plants of each expression clone.

Click here for additional data file.

Supplemental Information 3 Construction of the plasmid PJG090, the template for PCR amplification of two spacers.

Overlapping PCR was performed to produce the sequence of sgRNA–OsU6a promoter. The PCR product with attB adaptors was then introduced into pDONR207 with a BP reaction, producing the donor vector named PJG090, which is used as the template for PCR amplification of two spacers.

Click here for additional data file.

Supplemental Information 4 Optimizing the multiplex PCR to amplify a single sgRNA expression cassette in one tube.

(A) PJF997 digested by EcoRV was used as the template. OJD383 and OJD384 was used as spacer primers. (B) PJF999 digested by EcoRV was used as the template. OJG521 and OJG522 was used as spacer primers. Multiplex PCR reactions were set up with about 0.1 ng/ul of digested PJF997, universal primers (OJP001 and OJP002, 300 nm each) and spacer primers. To optimize the multiplex PCR, gradient concentration of spacer primers were used.

Click here for additional data file.

We thanked Prof. Li-Jia Qu (Peking University) and Prof. Yaoguang Liu (South China Agricultural University) for providing the plasmids of their CRISPR-Cas9 system.

Additional Information and Declarations

Competing Interests

Author Contributions

Data Availability

The authors declare that they have no competing interests.

Xiaoli Liu performed the experiments, analyzed the data, prepared figures and/or tables, authored or reviewed drafts of the paper, and approved the final draft.

Xiujuan Zhou performed the experiments, analyzed the data, prepared figures and/or tables, and approved the final draft.

Kang Li performed the experiments, prepared figures and/or tables, and approved the final draft.

Dehong Wang performed the experiments, authored or reviewed drafts of the paper, and approved the final draft.

Yuanhao Ding analyzed the data, authored or reviewed drafts of the paper, and approved the final draft.

Xianqing Liu performed the experiments, authored or reviewed drafts of the paper, and approved the final draft.

Jie Luo conceived and designed the experiments, authored or reviewed drafts of the paper, and approved the final draft.

Chuanying Fang conceived and designed the experiments, analyzed the data, authored or reviewed drafts of the paper, and approved the final draft.

The following information was supplied regarding data availability:

The raw data is shown in the figures and the tables.

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
