# Peer review of "A simple and efficient cloning system for CRISPR/Cas9-mediated genome editing in rice"

_PeerJ, doi:10.7717/peerj.8491_

## Round 0.1 · original submission · Major Revisions

The authors established a simple and efficient method to generate CRIPSR/Cas9 vectors, which is interesting improvement for genome editing research. However, substantial revisions on methodology description, data analysis, writing standard and figure presentation are absolutely necessary.

Reviewer 1 ·

Basic reporting

- the paper was written correctly, although minor language errors are present, such as "second round of PCR to producing" should be "to produce".
- the introduction should focus more on the topic of research and not on CRISPR technique and achievements in general. There is a lack of information about what other methods are used to obtain plasmids
- the results contain a lot of information that is repeated from the methods
- there is no comparison as to how this method compares with others
- there is no information in the literature regarding
Arite et al. 2007;
Arite et al. 2009
Khan et al. 2017
- In turn, Symington LS &Gautier J (2011) does not appear in the manuscript
- in the manuscript table s2 is before table s1
- no restriction sites are marked in table s2 (Line 111)
- Fig 3 i 4:
colony PCR rather than clone PCR
incubation and plasmid extraction rather than palsmid
Transformation into….
- Fig 5 a and b
more clearly mark the editing places and PAM
- Fig 5d sign the size marker bands and add the product from the control plant

Experimental design

There is a lack of details in the methodology so that the presented technique can be easily used in one's own research. The purpose of this research is to propose an alternative method to others, so the more it should be very clear and leave no doubt.

Validity of the findings

no comment

Additional comments

no comment

Reviewer 2 ·

Basic reporting

no comment

Experimental design

no comment

Validity of the findings

no comment

Additional comments

Technology involving the targeted mutagenesis of plants using programmable nucleases has been developing rapidly and has enormous potential in next-generation plant breeding. Notably, the clustered regularly interspaced short palindromic repeats/Cas9 (CRISPR/Cas9) system has paved the way for the development of rapid and cost-effective procedures to create new mutant populations in plants. Genome-edited plants from multiple species have been produced successfully using a method in which a Cas9-guide RNA (gRNA) expression cassette and selectable marker are integrated into the genomic DNA via Agrobacterium tumefaciens-mediated transformation or particle bombardment. Although CRISPR/Cas9 integration increases the chance of off-target modifications and foreign DNA sequences cause legislative concerns about genetically modified organisms, transformation of plants with Cas9-guide RNA (gRNA) expression cassette DNA is most popular and widely used approach for making genome-edited plants.
In the manuscript (peerj-42378), the authors established simple and efficient system for constructing CRISPR/Cas9-gRNA expression vectors to produce genome-edited rice plants. In this system, overlapping PCR and Gateway/Golden Gate system were successfully used for reducing the step and time for vector construction. The reviewer considers that present CRISPR/Cas9-gRNA expression vector construction system is highly useful to reduce molecular lab works such as PCR, subcloning, ligation. However, the reviewer thinks that the manuscript is too preliminary to be published in the The PeerJ.

The reviewer thinks that readers cannot understand what the authors conducted for producing new donor and destination vectors even if they carefully read Materials and Methods section. To avoid this, detailed flowcharts for the procedures of making donor and destination vectors should be presented as Figures (or Supplementary Figures), and please refer these figures in Materials and Methods section. Although showing several figures in Materials and Methods section is not so usual in scientific journal, it is not problematic but will be recommended for the present manuscript because contents of this study are mostly protocol-like for efficient producing of the CRISPR/Cas9-gRNA expression vector. Although present Figures 1 and 2 are flowchart-like, however, these figures are too rough to understand precise procedure. Figures 3 and 4 are also just outline, and important technical information are mostly omitted.

Lines 271-273: Please add descriptions for the estimated size (bp size) of long deletion of inter-region between targets 1 and 2 in OsRCD gene in the text, and discuss whether the estimated size fits with the experimentally detected deletion size in Figure 5D, or not.

Figure 5A and B: Sanger sequencing chromatograms for control and mutants should presented as top-bottom set (aligned) to clearly show the mutated position. In addition, please add information for the exact insertion/deletion regions like as “+1, A”, “-2 GG)” and etc.

Figure 5D: Information of base pair length for maker DNA is essential to estimate the size of deletion.

Supplementary Table 1: Please show number of plants that possessed both editions/mutations in targets 1 and 2 of OsRCD gene.

---

## Round 0.2 · accepted · Accept

CRISPR-Cas9 based genome editing has become the method of choice for the rapid generation of transgenic plants. We hope that the fast and efficient cloning system the authors presented could facilitate genetic engineering in plants.

Reviewer 2 ·

Basic reporting

Technology involving the targeted mutagenesis of plants using programmable nucleases has been developing rapidly and has enormous potential in next-generation plant breeding. Notably, the clustered regularly interspaced short palindromic repeats/Cas9 (CRISPR/Cas9) system has paved the way for the development of rapid and cost-effective procedures to create new mutant populations in plants. Genome-edited plants from multiple species have been produced successfully using a method in which a Cas9-guide RNA (gRNA) expression cassette and selectable marker are integrated into the genomic DNA via Agrobacterium tumefaciens-mediated transformation or particle bombardment. Although CRISPR/Cas9 integration increases the chance of off-target modifications and foreign DNA sequences cause legislative concerns about genetically modified organisms, transformation of plants with Cas9-guide RNA (gRNA) expression cassette DNA is most popular and widely used approach for making genome-edited plants.

In the manuscript (peerj-42378), the authors established simple and efficient system for constructing CRISPR/Cas9-gRNA expression vectors to produce genome-edited rice plants. In this system, overlapping PCR and Gateway/Golden Gate system were successfully used for reducing the step and time for vector construction. The reviewer considers that present CRISPR/Cas9-gRNA expression vector construction system is highly useful to reduce molecular lab works such as PCR, subcloning, ligation. Although figures for the former version of the manuscript were too preliminary to be published in the The PeerJ, the figures and descriptions for these figures were largely improved in the present revised manuscript. Therefore, the manuscript will be acceptable for the publication in The PeerJ.

Experimental design

OK

Validity of the findings

OK

Additional comments

All concerns indicated by the reviewer indicated were correctly revised in the manuscript. Therefore, the reviewer thinks that the manuscript is now suitable for publication in The PeerJ.